# Oriented Object Detection in Remote Sensing Images with Anchor-Free Oriented Region Proposal Network

Jianxiang Li [ID], Yan Tian *[ID], Yiping Xu [ID] and Zili Zhang [ID]

School of Electronic Information and Communications, Huazhong University of Science and Technology, Wuhan 430074, China; jianxiang_li@hust.edu.cn (J.L.); xuyiping@hust.edu.cn (Y.X.); zhangzili@hust.edu.cn (Z.Z.)
* Correspondence: tianyan@hust.edu.cn

**Abstract:** Oriented object detection is a fundamental and challenging task in remote sensing image analysis that has recently drawn much attention. Currently, mainstream oriented object detectors are based on densely placed predefined anchors. However, the high number of anchors aggravates the positive and negative sample imbalance problem, which may lead to duplicate detections or missed detections. To address the problem, this paper proposes a novel anchor-free two-stage oriented object detector. We propose the Anchor-Free Oriented Region Proposal Network (AFO-RPN) to generate high-quality oriented proposals without enormous predefined anchors. To deal with rotation problems, we also propose a new representation of an oriented box based on a polar coordinate system. To solve the severe appearance ambiguity problems faced by anchor-free methods, we use a Criss-Cross Attention Feature Pyramid Network (CCA-FPN) to exploit the contextual information of each pixel and its neighbors in order to enhance the feature representation. Extensive experiments on three public remote sensing benchmarks—DOTA, DIOR-R, and HRSC2016—demonstrate that our method can achieve very promising detection performance, with a mean average precision (mAP) of 80.68%, 67.15%, and 90.45%, respectively, on the benchmarks.

**Keywords:** remote sensing images; oriented object detection; contextual information; Anchor Free Region Proposal Network; polar representation

## 1. Introduction

Object detection is a fundamental and challenging task in computer vision. Object detection in remote sensing images (RSIs) [1–9], which recognizes and locates the objects of interest such as vehicles [4,5], ships [6,7], and airplanes [8,9] on the ground, has enabled applications in fields such as traffic planning and land surveying.

Traditional object detection methods [10], like object-based image analysis (OBIA) [11], usually take two steps to accomplish object detection: firstly, extract regions that may contain potential objects, then extract hand-designed features and apply classifiers to obtain the class information. However, their detection performance is unsatisfactory because the handcrafted features have limited representational power with insufficient semantic information.

Benefitting from the rapid development of deep convolutional neural networks (DC-NNs) [12] and publicly available large-scale benchmarks, generic object detection [13–19] has made extensive progress in natural scenes, which has also prompted the increased development of object detection in RSIs. Generic object detectors employ an axis-aligned bounding box, also called a horizontal bounding box (HBB), to localize the object in the image. However, detecting objects in RSIs with HBBs remains a challenge. Because RSIs are photographed from a bird's eye view, the objects in RSIs often have large aspect ratios and dense arrangements, as is the case with, for example, ships docked in a harbor. As a result, oriented bounding box (OBB) has recently been adopted to describe the position of the arbitrary-rotated object in RSIs.

Currently, mainstream oriented object detectors [20–23] are based on densely placed predefined anchors. Several early rotation detectors use a horizontal anchor-based Region Proposal Network (RPN) to generate horizontal regions of interest (RoIs), and then design novel network modules to convert the horizontal RoIs into OBBs. For example, Ding et al. [20] build a rotated RoI learner to transform horizontal RoIs into rotated RoIs (RRoIs), and then regress the RRoIs to obtain the final results. However, the horizontal RoI typically contains massive ground pixels and other objects due to the arbitrary orientation and dense distribution of the objects, as shown in Figure 1a. The mismatch between the horizontal anchors and rotation objects causes difficulties in network training and further degrades performance [21].

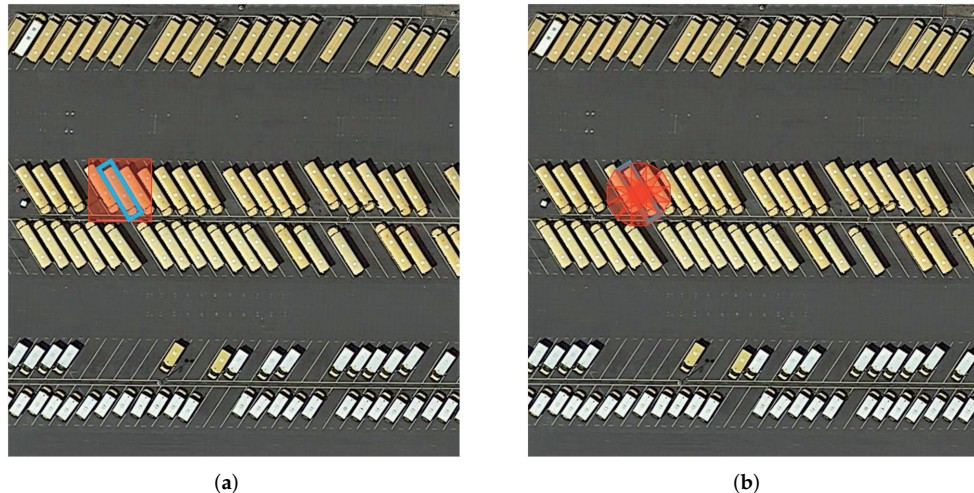

(a)　　　　　　　　　　　　　　　　　　　　(b)

**Figure 1.** Disadvantages of anchor-based detectors. The blue rectangle represents the ground truth, and the orange rectangle represents the anchor box. (**a**) The horizontal anchor contains massive ground pixels and other objects. (**b**) RRPN often places too many oriented anchors to ensure a high recall rate.

To address the problem, some detectors use a rotated anchor-based RPN (RRPN) [23] to generate RRoIs. Nevertheless, the Intersection over Union (IoU) is highly sensitive to the angle. To ensure the high recall rate, RRPN places 54 rotated anchors (six orientations, three aspect ratios, and three scales) for each sample point on the feature map, as shown in Figure 1b. However, the high number of anchors increases the computational burden and aggravates the imbalance between positive and negative samples. Moreover, dense anchors may lead to duplicate detections of the same object and missed detections [21] after the non-maximum suppression (NMS).

Owing to the above problems, the use of anchor-free oriented object detectors is increasing. Anchor-free detectors directly locate the objects without manually defined anchors. In particular, keypoint-based methods use several points, such as corners [24], extreme points [25], and the center [26], to represent the positive samples and directly regress the categories and locations of the objects from the features of the keypoints. For example, CenterNet [26] uses one center point to represent the object and directly regresses other properties, such as object size, dimension, and pose, from the features at the center position. Most anchor-free oriented object detectors are inherent from CenterNet for high efficiency and generality, having achieved performance competitive with anchor-based detectors. For example, Pan et al. [27] extend the CenterNet by adding a branch to regress the orientations of the OBBs, and the proposed DRN achieved consistent gains across multiple datasets in comparison with baseline approaches.

However, keypoint-based anchor-free object detectors face severe appearance ambiguity problems with backgrounds or other categories. As shown in Figure 2, the central areas of the objects are similar to the backgrounds, and some objects belonging to dif-

ferent categories even share the same center parts. The main reason for this is that the commonly used fully convolutional networks have insufficient contextual information [28] because of the limited local receptive fields due to fixed DCNN structures. Furthermore, nearly all anchor-free detectors are one-stage detectors, which usually encounter severe misalignment [29] between the axis-aligned convolutional features extracted by the DCNNs and rotational bounding boxes. However, the feature warping module of the two-stage detectors, such as RRoI Pooling [23] or RRoI Align [20], can alleviate this problem.

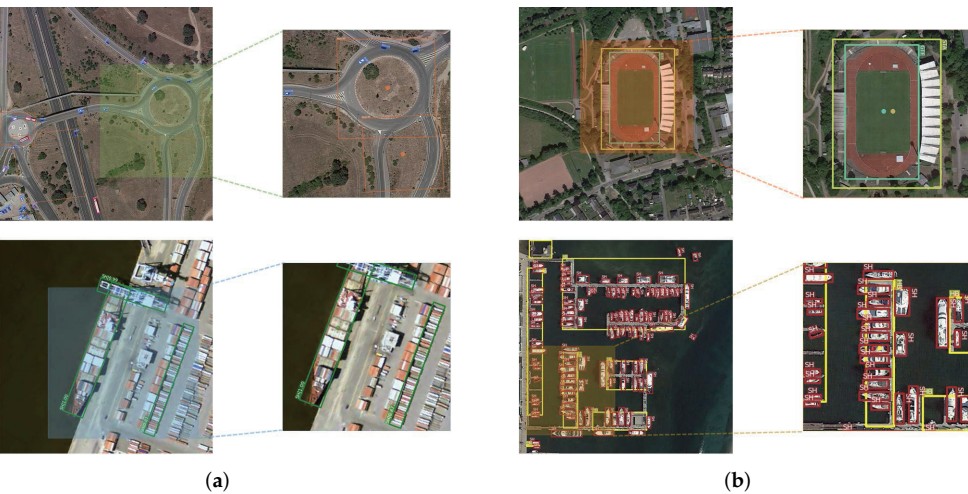

(**a**)                 (**b**)

**Figure 2.** Appearance ambiguity problems of the keypoint-based anchor-free object detectors. (**a**) The central areas of the objects are similar to the backgrounds. (**b**) Some different categories objects share the same center parts.

Based on the above discussion, we propose a novel two-stage oriented object detector, following the coarse- to fine-detection paradigm. Our method consists of four components: a backbone, a Criss-Cross Attention Feature Pyramid Network (CCA-FPN), an Anchor-Free Oriented Region Proposal Network (AFO-RPN) and oriented RCNN heads.

At the outset, we use the proposed AFO-RPN to generate high quality–oriented proposals without placing excessive fix-shaped anchors on the feature map. To enhance the feature representation of each pixel in the feature map, we adopt CCA-FPN to exploit the contextual information from full image patch. To deal with rotation problems, we propose a new representation of OBB based on polar coordinate system. Finally, we apply an AlignConv to align the features and then use oriented RCNN heads to predict the classification scores and regress the final OBBs. To demonstrate the effectiveness of our method, we conducted extensive experiments on three public RSI oriented object detection datasets—DOTA [30], DIOR-R [31], and HRSC2016 [7].

The contributions of this paper can be summarized as follows: (1) We propose a new anchor-free oriented object detector following the two-stage coarse-to-refined detection paradigm. Specifically, we proposed AFO-RPN to generate high-quality proposals without enormous predefined anchors and a new representation method of OBB in the polar coordinate system, which can better handle the rotation problem; (2) We apply CCA module into FPN to enhance the feature representation of each pixel by capturing the contextual information from the full patch image; and (3) Experimental results on three publicly available datasets show that our method achieves promising results and outperforms previous state-of-the-art methods.

The rest of this paper is organized as follows. Section 2 reviews the related work and explains our method in details. Section 3 compares the propsed method with state-of-the-art methods on different datasets. Section 4 discusses the ablation experiments of the proposed method. Section 5 offers our conclusions.

## 2. Materials and Methods

### 2.1. Related Work

#### 2.1.1. Generic Object Detection

With recent advances in deep learning techniques, the performance of DCNN-based generic object detectors has improved significantly. Generic object detectors aim to detect general objects in natural scenes with HBBs to locate objects. The mainstream generic object detection methods can be roughly divided according to the following standards: two- or single-stage object detection, and anchor-free or anchor-based object detection.

Two-stage object detectors, such as Faster RCNN [13] and Mask RCNN [14], first generate RoIs, which can be treated as coarse class-agnostic detection results, and then in the second stage extract the RoI features to perform refined classification and location. Two-stage object detectors can achieve high detection accuracy, but their inference speed is slow. One-stage object detectors, such as YOLO series [15–17], SSD [18], and RetinaNet [19], directly regress the complete detection results through one-step prediction. One-stage detectors are fast and can achieve real-time inference, but they are less accurate than two-stage detectors. The design of anchors has been popularized by Faster R-CNN in its RPN and has become the convention in many modern object detectors.

Although anchor-based detectors currently dominate in the object detection arena, they involve placing a dense set of predefined anchors at each location of the feature map, which dramatically increases the computational cost. As a result, anchor-free detectors [24–26,32–34], which directly locate the object without manually defined anchors, have become popular. For example, CornerNet [24] directly regresses the top-left and bottom-right corner points and then groups them to form the final HBB. ExtremeNet [25] predicts four extreme points (top-most, left-most, right-most, and bottom-most) and one center point, and then groups them into the HBB. CenterNet [26] models an object as one single point and directly regresses the center point of the HBB. Unlike key point-based anchor-free detectors, which treat several key points of the objects as positive samples, pixel-based anchor-free detectors attempt to solve the problem in a per-pixel prediction fashion. RepPoints [32] introduces a set of representative points that adaptively learn to position themselves over an object. Tian et al. [33] regard all the pixels inside the object HBB as positive samples. Motivated by the human eye system, Kong et al. [34] regard the pixels inside the fovea area of the object HBB as the positive samples. Both of them predict four distances to the four sides of HBB from the positive pixels to form the HBB. Anchor-free detection methods are fast in inference and also achieve competitive detection results with anchor-based detection methods.

#### 2.1.2. Oriented Object Detection

Oriented object detection is receiving significant attention in areas such as remote sensing images and natural scene text. Oriented object detectors use OBBs to locate arbitrary-rotated objects other than HBBs because the objects in those scenes usually have large aspect ratios and are densely packed.

Oriented object detectors often use generic object detectors as a baseline and then add specially designed modules to regress OBB from HBB. Based on Faster-RCNN [13], RRPN [23] uses Rotation RPN and Rotation RoI pooling for arbitrary-oriented text detection. The RoI Transformer [20] utilizes a learnable module to transform horizontal RoIs to RRoIs. Xu et al. [22] propose to glide each vertex on the four corresponding sides of HBB to represent OBB, and Ye et al. [35] introduce feature fusion and feature filtration modules to exploit multilevel context information.

Based on RetinaNet [19], ADT-Det [36] uses a feature pyramid transformer that enhances features through feature interaction with multiple scales and layers. S$^2$A-Net [29] utilizes a feature alignment module for full feature alignment and an oriented detection module to alleviate the inconsistency between classification and regression. R$^3$Det [37] uses a feature refinement module to re-encode the position information and then reconstruct the entire feature map through pixel-wise interpolation.

Some research has adopted the OBB based on the semantic-segmentation network, such as Mask RCNN [14]. Mask OBB [38] is the first to treat the oriented object detection as an instance segmentation problem. Wang et al. [39] propose a center probability map OBB that gives a better OBB representation by reducing the influence of background pixels inside the OBB and obtaining higher detection performance.

Aside from the above anchor-based detectors, some rotation object detectors use an anchor-free approach. Based on CenterNet [26], Pan et al. [27] propose DRN by adding a branch to regress the orientations of the OBBs, and Shi et al. [40] develop a multi-task learning procedure to weight multi-task loss function during training. Other anchor-free detectors use new OBB representations. Xiao et al. [41] adopt FCOS [33] as the baseline and propose axis learning to detect oriented objects by predicting the axis of the object. Guo et al. [42] propose CFA, which uses RepPoints [32] as its baseline, and construct a convex-hull set for each oriented object.

### 2.1.3. Contextual Information and Attention Mechanisms

Numerous studies have shown that using contextual information and attention mechanisms can improve the performance of vision tasks such as scene classification, object detection, and instance segmentation.

For example, Wang et al. [43] use a novel locality and structure regularized low-rank representation method to characterize the global and local structures for hyper-spectral image classification task. ARCnet [44] utilizes a novel recurrent attention structure to force the scene classifiers to learn to focus on some critical areas of the very high-resolution RSIs, which often contain complex objects. AGMFA-Net [45] uses an attention-guided multi-layer feature aggregation network to capture more complete semantic regions for more powerful scene representation.

Contextual information aggregation has been widely adopted in semantic segmentation networks. To enhance the ability of the network to distinguish small-scale objects, CFEM [46] uses a context-based feature enhancement module to enhance the discriminant ability to distinguish small objects. HRCNet [47] utilizes a lightweight high-resolution context extraction network to acquire global context information and recognize the boundary being.

The usefulness of contextual information has been verified by many studies [35,48,49] in aerial object detection, especially when object appearances are insufficient due to small size, occlusion, or complicated backgrounds. CADNet [48] incorporates global and local contextual information and has a spatial-and-scale awareness attention module for object detection in RSIs. Wu et al. [49] propose a local context module that establishes the positional relationships between a proposal and its surrounding region pixels to help detect objects. $\mathcal{F}$3-Net [35] uses a feature fusion module that extracts the contextual information at different scales.

Attention mechanisms also show promise in oriented object detection by guiding the processing to more informative and relevant regions. ROSD [50] uses an orientation attention module to enhance the orientation sensitivity for accurate rotated object regression. CFC-Net [51] utilizes polarized attention to construct task-specific critical features. Li et al. [52] use a center-boundary dual attention module to extract attention features on the oriented objects' center and boundary regions. RADet [53] uses a multi-layer attention network focused simultaneously on objects' spatial position and features. SCRDet [54] uses a supervised multi-dimensional attention network consisting of a pixel attention network and channel attention network to suppress the noise and highlight the foreground.

### 2.1.4. OBB Representation Methods

The two most widely used OBB representation methods are the angle-based five-parameter representation method and the vertex-based eight-parameter representation method. The more commonly used five-parameter representation directly adds an angle parameter $\theta$ to HBB representation $(x, y, w, h)$, and the definition of the angle $\theta$ is the

acute angle determined by the long side of the rectangle and X-axis. The eight-parameter representation directly adopts the four corners of the OBB, e.g., $(x_1, y_1, x_2, y_2, x_3, y_3, x_4, y_4)$.

Although oriented object detectors with either form of OBB representation have demonstrated good performance, the inherent drawbacks of these two representations hinder the further improvement of the detection results [55]. The angular parameters embedded in the five-parameter representation encounter the problem of angular periodicity, leading to difficulty in the learning process. In contrast, the eight-parameter representation requires the exact same points order of ground truth and prediction, which otherwise leads to an unstable training process.

To handle these problems, some detectors have introduced new representations along with the anchor-free model. Axis learning [41] locates objects by predicting their axis and width, the latter of which is vertical to the axis. O²DNet [56] treats the objects as pairs of middle lines. SAR [57] uses a brand-new representation with a circle-cut horizontal rectangle. Wu et al. [58] propose a novel projection-based method for describing OBB. Yi et al. [59] propose BBAVectors to regress one center point and four middle points of the corresponding sides to form the OBB. X-LineNet [9] uses paired appearance-based intersecting line segments to represent aircraft.

The above representations are all based on cartesian coordinates, and recently, the representation based on polar coordinates has been employed for rotated object detection and instance segmentation. Polar Mask [60], which model instance masks in the polar coordinates as one center and n rays, achieves competitive performance with much simpler and more flexible. Polar coordinates-based representations have been proved helpful in rotation and direction-related problems. Following Polar Mask, some rotated object detectors [61,62] also adopt polar representation and show great potential. PolarDet [61] represents the OBB by multiple angles and shorter-polar diameter ratios. However, the OBB representation of PolarDet needs 13 parameters, and some of them are redundant. In contrast, we propose a similar but more efficient representation method with only seven parameters. P-RSDet [62] regresses three parameters in polar coordinates, which include a polar radius $\rho$ and the first two angles, to form the OBB and put forward a new Polar Ring Area Loss to improve the prediction accuracy.

### *2.2. Method*

#### 2.2.1. Overall Architecture

As shown in Figure 3, the proposed detector follows the two-stage detection paradigm, and contains four modules: the backbone for feature extraction, a CCA-FPN for feature representation enhancement with contextual information, an AFO-RPN for RRoI generation, and oriented RCNN heads for the final class and locations of the rotational object. For the backbone, we adopted ResNet [12], which is commonly used in many oriented detectors.

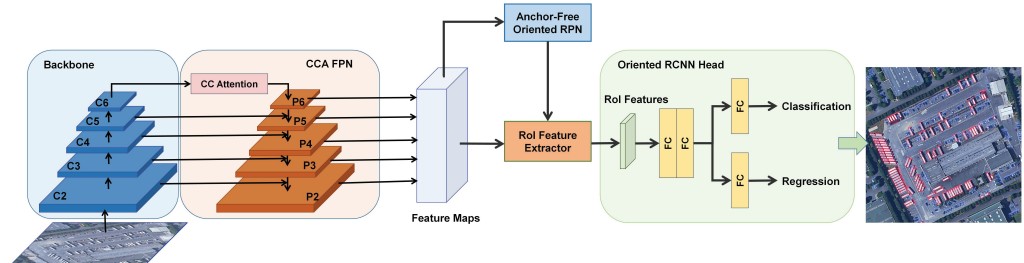

**Figure 3.** Overall architecture of the proposed method. There are four modules: backbone, Criss-Cross Attention FPN, anchor-free oriented RPN, and oriented RCNN heads.

#### 2.2.2. Criss-Cross Attention FPN

Contextual information has been shown to be helpful in many computer vision tasks, such as scene classification, object detection, and semantic segmentation. In general,

contextual information in vision describes the relationship between a pixel and its surrounding pixels.

One of the characteristics of RSIs is that the same category objects are often distributed in a particular region, such as vehicles in a parking lot or ships in a harbor. Another characteristic is that objects are closely related to the scene—for example, airplanes are closely related to an airport, and ships are closely related to the water.

Motivated by the above observations and analysis, we propose a Criss-Cross Attention FPN to fully exploit the contextual information of each pixel and its neighbors, which enhances the feature representation of the objects. Specifically, we embed the cascaded criss-cross attention modules into the FPN to enhance the pixel representations. The criss-cross attention module first used in CC-Net [28] is designed to collect the contextual information in the criss-cross path in order to enhance the pixel representative ability by modeling full-patch image dependencies over local features.

Given a feature map $\mathbf{H} \in \mathbb{R}^{C \times W \times H}$, we first apply three $1 \times 1$ convolutional layers on $\mathbf{H}$ to obtain three feature maps: queries map $\mathbf{Q}$, keys map $\mathbf{K}$, and values map $\mathbf{V}$. Note that $\mathbf{Q}$ and $\mathbf{K}$ have the same dimension, where $\{\mathbf{Q}, \mathbf{K}\} \in \mathbb{R}^{C' \times W \times H}$, and $\mathbf{V}$ has the same dimension as $\mathbf{H}$. We set $C'$ less than $C$ for the purpose of dimension reduction.

Next, we obtain a vector $\mathbf{Q_u}$ at each spatial position $\mathbf{u}$ of $\mathbf{Q}$ and the set $\mathbf{\Omega_u}$ in which the vectors are extracted from the same row and column with spatial position $\mathbf{u}$ from keys map $\mathbf{K}$. The correlation vector $\mathbf{D_u}$ is calculated by applying affinity operation on query vector $\mathbf{Q_u}$ and key vector set $\mathbf{\Omega_u}$ as follows:

$$\mathbf{D_u} = \mathbf{Q_u}\mathbf{\Omega_u}^T, \tag{1}$$

where $\mathbf{D_u} \in \mathbb{R}^{W+H-1}$. Next, we calculate the attention vector $\mathbf{A_u}$ by applying softmax function on $\mathbf{D_u}$ over the channel dimension, as follows:

$$\mathbf{A_u} = softmax(\mathbf{D_u}). \tag{2}$$

Then, we obtain the value vector set $\mathbf{\Phi_u}$, in which the value vectors are extracted from the same row and column with position $\mathbf{u}$ of $\mathbf{V}$. The contextual information is collected by an aggregation operation defined as:

$$\mathbf{H'_u} = \sum_{i=0}^{W+H-1} \mathbf{A}_{i,\mathbf{u}}\mathbf{\Phi}_{i,\mathbf{u}} + \mathbf{H_u}, \tag{3}$$

where $\mathbf{H'} \in \mathbb{R}^{C \times W \times H}$ is the output of criss-cross attention module, which aggregates contextual information together with each pixel. A single criss-cross attention module can only capture contextual information of pixels in horizontal and vertical directions. However, it is not sufficient to focus only on the criss-cross path information for the problem of oriented object detection. To capture the contextual information in other directions, we use two cascaded criss-cross attention modules, following CC-Net [28].

2.2.3. Anchor-Free Oriented Region Proposal Network

As shown in Figure 3, the CCA-FPN produces five levels of feature maps $\{P_2, P_3, P_4, P_5, P_6\}$, where their strides $\{s_2, s_3, s_4, s_5, s_6\}$ are 4, 8, 16, 32, and 64, respectively. The proposed AFO-RPN takes the feature map $P_i$ as input and outputs a set of oriented proposals, as shown in Figure 4. We introduce the polar representation method of OBB and then present the details of AFO-RPN.

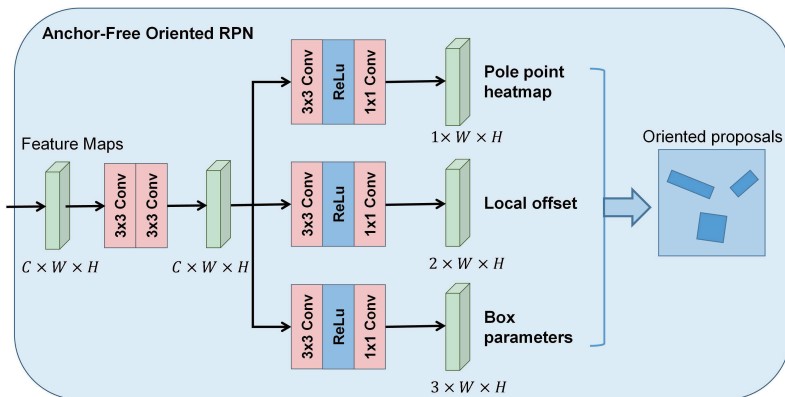

**Figure 4.** Details of the proposed AFO-RPN.

### 2.2.4. Polar Representation of OBB

Instead of the commonly used Cartesian-based OBB representation, we use the polar-based OBB representation in this paper, as shown in Figure 5. Specifically, the centroid of each object is used as the origin of the polar coordinates, and we use $(c_x, c_y, \rho, \gamma, \varphi)$ to represent the OBB, where $c_x, c_y$ are the centroids of the OBB, which are also the poles of the polar coordinates. $\rho$ is the radius, which calculates the distance from the centroid to the vertex, and $\gamma$ is the central angle corresponding to the short side of the OBB. This representation is more robust than the one that uses $w$ and $h$ to represent a rectangular box's long and short sides. The reason is that using $w$ and $h$ to represent the rectangular box is prone to the problem of confusion between $w$ and $h$ when the rectangular box is close to the square [55]. However, by using $\rho$ and $\gamma$ to represent rectangular, the confusion between $w$ and $h$ can be avoided. $\varphi$ represents the rotation angle of the OBB, which is defined in the polar coordinate system. We define the beginning angle $0°$ to coincide with the positive y-axis and increase the angle counterclockwise.

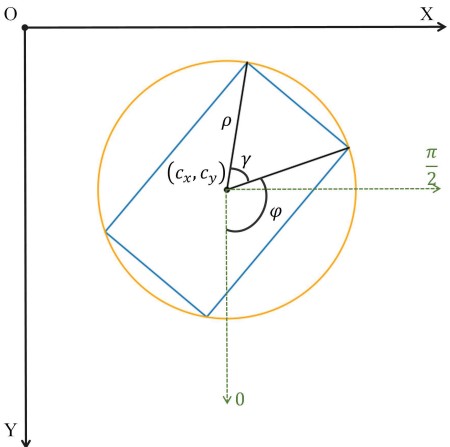

**Figure 5.** Proposed polar representation of OBB.

### 2.2.5. Pole Point Regression

Following previous work such as CenterNet [26], we use pole point (center point of the OBB) heatmaps to represent the location and objectness of the objects. Unlike CenterNet, which uses a 2D Gaussian kernel with a diagonal correlation matrix to map the key point to heatmaps, we use the rotated Gaussian kernel with a correlation matrix related to the rotation angle of the ground truth box.

Specifically, for an OBB ground truth $(c_x, c_y, w, h, \theta)$, we place a 2D Gaussian distribution $\mathcal{N}(\mathbf{m}, \boldsymbol{\Sigma})$ to form the ground truth heatmap in the training stage. Here, $\mathbf{m} = \left( \left\lfloor \frac{c_x}{s} \right\rfloor, \left\lfloor \frac{c_y}{s} \right\rfloor \right)$

represents the center of the gaussian distribution mapped into the feature map, where $s$ is the downsampling stride of each feature map. The correlation matrix $\boldsymbol{\Sigma}$ is calculated as:

$$\boldsymbol{\Sigma}^{\frac{1}{2}} = \mathbf{R}(\theta)\mathbf{S}\mathbf{R}^T(\theta), \tag{4}$$

where the rotation matrix $\mathbf{R}(\theta)$ is defined as:

$$\mathbf{R}(\theta) = \begin{bmatrix} \cos\theta & -\sin\theta \\ \sin\theta & \cos\theta \end{bmatrix}. \tag{5}$$

$\mathbf{S} = diag(\sigma_x, \sigma_y)$ is the standard deviation matrix, where $\sigma_x = w \times \sigma_p$, $\sigma_y = h \times \sigma_p$, and $\sigma_p$ is an object size-adaptive standard deviation [26].

In the training stage, only the peaks of the Gaussians are treated as the positive samples; all the other points are negative. To handle the imbalance between the positive and negative samples, we use a pixel-wise logistic regression with variant focal loss as CenterNet [26]:

$$\mathcal{L}_k = \frac{-1}{N} \sum_{xy} \begin{cases} (1 - \hat{Y}_{xy})^\alpha \log(\hat{Y}_{xy}), & \text{if } \hat{Y}_{xy} = 1 \\ (1 - Y_{xy})^\beta \log(\hat{Y}_{xy})^\alpha \log(1 - \hat{Y}_{xy}), & \text{otherwise} \end{cases}, \tag{6}$$

where $\hat{Y}_{xy}$ and $Y_{xy}$ refer to the ground-truth and the predicted heatmap values, $\alpha$ and $\beta$ are the hyper-parameters of the focal loss that control the contribution of each point, and $N$ is the number of the objects in the input image.

Furthermore, to compensate for the quantization error caused by the output stride, we additionally predict a local offset map $\mathbf{O} \in \mathbb{R}^{2 \times H \times W}$, slightly adjust the center point locations before remapping them to the input resolution, and the offset of the OBB center point is defined as $\mathbf{o} = \left( \frac{c_x}{s} - \lfloor \frac{c_x}{s} \rfloor, \frac{c_y}{s} - \lfloor \frac{c_y}{s} \rfloor \right)$.

The offset is optimized with a smooth $L_1$ loss [13]:

$$\mathcal{L}_{\mathbf{O}} = \frac{1}{N} \sum_k Smooth_{L_1}(\mathbf{o}_k - \hat{\mathbf{o}}_k), \tag{7}$$

where $\acute{\mathbf{o}}_k$ and $\mathbf{o}_k$ refer to the ground-truth and the predicted local offset of the $k$th object, respectively. The smooth $L_1$ loss is defined as:

$$Smooth_{L_1} = \begin{cases} 0.5x^2, & \text{if } |x| < 1 \\ |x| - 0.5, & \text{otherwise} \end{cases}. \tag{8}$$

### 2.2.6. Box Parameters Regression

The box parameters are defined as $\mathbf{b} = (\rho, \gamma, \varphi)$, where $\rho$ is the radius that calculates the distance from the centroid to the vertex, $\gamma$ is the central angle corresponding to the short side of the OBB, and $\varphi$ represents the rotation angle of the OBB, as depicted in Figure 5. We predict the box parameter map $\mathbf{B} \in \mathbb{R}^{3 \times W \times H}$ with a smooth L1 loss:

$$\mathcal{L}_{\mathbf{B}} = \frac{1}{N} \sum_k Smooth_{L_1}\left(\mathbf{b}_k - \hat{\mathbf{b}}_k\right), \tag{9}$$

where $\hat{\mathbf{b}}_k$ and $\mathbf{b}_k$ refer to the ground truth and the predicted box parameters of the $k$th object, respectively.

The overall training loss of AFO-RPN is:

$$\mathcal{L}_{AFO-RPN} = \mathcal{L}_k + \lambda_O \mathcal{L}_{\mathbf{O}} + \lambda_B \mathcal{L}_{\mathbf{B}}, \tag{10}$$

where $\lambda_O$ and $\lambda_B$ are the weighted factors to control the contributions of each item, and we set $\lambda_O = 1$ and $\lambda_B = 0.1$ in our experiments.

### 2.2.7. Oriented RCNN Heads

As shown in Figure 6, the RoI feature extractor takes a group of feature maps {P2, P3, P4, P5, P6} and a set of oriented proposals as input. We use the align conv module to extract a fix-sized RoI feature from the corresponding feature map. The details of the align conv can be referred to $S^2$A-Net [29]. Then we use two fully connected layers and two sibling fully connected layers to predict the classification scores and regress the final oriented bounding boxes, as shown in Figure 3. The loss of RCNN heads is the same as that in [20]. The RCNN heads loss is given by:

$$\mathcal{L}_{head} = \frac{1}{N_{cls}} \sum_i \mathcal{L}_{cls} + \frac{1}{N_{reg}} \sum_i p_i^\star \mathcal{L}_{reg}, \tag{11}$$

where $N_{cls}$ and $N_{reg}$ are the number of proposals generated by AFO-RPN and the positive proposals in a mini batch, respectively. $p_i^\star$ is an index and when $i$th proposal is positive, it is 1, otherwise it is 0.

The total loss function of the proposed method follows the multitask learning way, and it is defined as:

$$\mathcal{L}_{total} = \lambda_{AFO-RPN}\mathcal{L}_{AFO-RPN} + \lambda_{head}\mathcal{L}_{head}, \tag{12}$$

where $\lambda_{AFO-RPN}$ and $\lambda_{head}$ are the weighted factors, and we set $\lambda_{AFO-RPN} = 1$ and $\lambda_{head} = 1$.

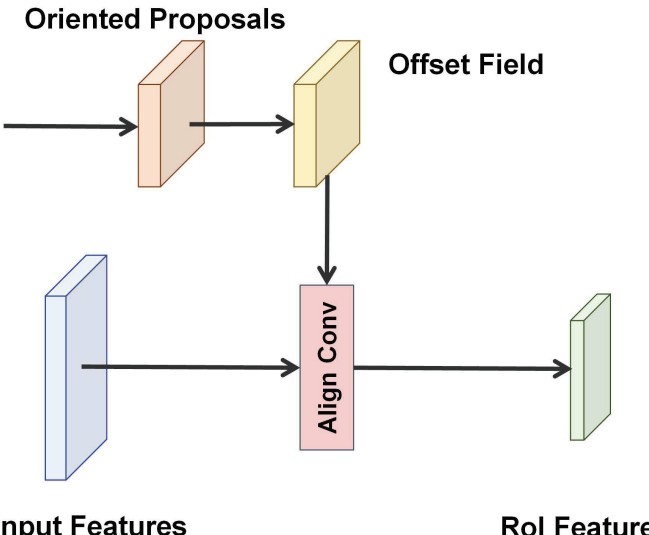

**Figure 6.** The details of RoI feature extractor module.

## 3. Results

### 3.1. Datasets

#### 3.1.1. DOTA

DOTA [30] is one of the largest public aerial image detection datasets. It contains 2806 images ranging from $800 \times 800$ to $4000 \times 4000$ pixels and 188,282 instances labeled by arbitrarily oriented quadrilaterals over 15 categories: plane (PL), baseball diamond (BD), bridge (BR), ground track field (GTF), small vehicle (SV), large vehicle (LV), ship (SH), tennis court (TC), basketball court (BC), storage tank (ST), soccer-ball field (SBF), roundabout (RA), harbor (HA), swimming pool (SP), and helicopter (HC). The total dataset is divided into the training set (1411 images), validation set (458 images), and test set (937 images). We used the training set for network training and the validation set for evaluation in the ablation experiments. In a comparison with state-of-the-art object detectors, the training set and validation set were both used for network training, and the corresponding results on the

test set were submitted to the official evaluation server at https://captain-whu.github.io/DOTA/evaluation.html (accessed on 27 January 2022). Following [20], we crop the original images into $1024 \times 1024$ patches with a stride 200 for training and testing. For multi-scale training and testing, we first resize original images at three scales (0.5, 1.0, and 1.5) which are chosen empirically, and then crop them into $1024 \times 1024$ patches with a stride of 512.

### 3.1.2. DIOR-R

DIOR-R [31] is a revised dataset of DIOR [1], which is another publicly available arbitrary-oriented object detection dataset in the earth observation community. It contains 23,463 images with a fixed size of $800 \times 800$ pixels and 192,518 annotated instances, covering a wide range of scenes. The spatial resolutions range from 0.5 m to 30 m. The objects of this dataset belong to 20 categories: airplane (APL), airport (APO), baseball field (BF), basketball court (BC), bridge (BR), chimney (CH), expressway service area (ESA), expressway toll station (ETS), dam (DAM), golf field (GF), ground track field (GTF), harbor (HA), overpass (OP), ship (SH), stadium (STA), storage tank (STO), tennis court (TC), train station (TS), vehicle (VE), and windmill (WM). The dataset is divided into the training (5862 images), validation (5863 images), and test (11,738 images) sets. For a fair comparison with other methods, the proposed detector is trained on the train+val set and evaluated on the test set.

### 3.1.3. HRSC2016

HRSC2016 [7] is an oriented ship detection dataset that contains 1061 images of rotated ships with large aspect ratios, collected from six famous harbors, including ships on the sea and close in-shore. The images range from $300 \times 300$ to $1500 \times 900$ pixels, and the ground sample distances are between 2 m and 0.4 m. The dataset is randomly split into the training set, validation set, and test set, containing 436 images including 1207 instances, 181 images including 541 instances, and 444 images including 1228 instances, respectively. We used both the training and validation sets for training and the test set for evaluation in our experiments. All images were resized to $800 \times 1333$ without changing the aspect ratio.

### 3.2. Implementation Details

We used ResNet 101 [12] as the backbone network for comparisons with state-of-the-art methods. Our model was implemented on the mmdetection [20] library. We optimized the model by using the SGD algorithm, and the initial learning rate was set to 0.005. The momentum and weight decay were 0.9 and 0.0001, respectively. The DOTA and DIOR-R datasets were trained by 12 epochs in total, and the learning rate was divided by 10 at eight epochs and 11 epochs, respectively. The HRSC2016 dataset was trained by 36 epochs in total, and the decay steps were 24 and 33 epochs. We used one Nvidia Titan XP GPU for all the experiments.

In this article, we adopt the mean Average Precision (mAP) metric to evaluate the multi-class detection accuracy of all experiments. mAP is the average of AP values for all classes:

$$\mathrm{mAP} = \frac{\sum_{i=1}^{N} \mathrm{AP}_i}{N},$$ (13)

where $N$ is number of classes. The AP metric is measured by the area under the precision-recall curve. The higher the mAP value, the better the performance, and vice versa.

### 3.3. Comparisons with State-of-the-Art Methods

3.3.1. Results on DOTA

To validate the effectiveness of our method, we compared it with several state-of-the-art methods on the DOTA dataset test set. The results were evaluated by the official DOTA evaluation server. As shown in Table 1, our model achieved a 76.57% mAP, which is higher than many advanced methods. With the multi-scale training and testing strategy, our model achieved an 80.68% mAP. Some detection results are shown in Figure 7.

**Table 1.** Comparisons with state-of-the-art methods on DOTA dataset test set. * means multi-scale training and testing. **Bold** denotes the best detection results.

| Method | Backbone | PL | BD | BR | GTF | SV | LV | SH | TC | BC | ST | SBF | RA | HA | SP | HC | mAP(%) |
|---|---|---|---|---|---|---|---|---|---|---|---|---|---|---|---|---|---|
| One-stage |
| DAL [63] | ResNet 101 | 88.61 | 79.69 | 46.27 | 70.37 | 65.89 | 76.10 | 78.53 | 90.84 | 79.98 | 78.41 | 58.71 | 62.02 | 69.23 | 71.32 | 60.65 | 71.78 |
| ProjBB-R [58] | ResNet 101 | 88.96 | 79.32 | 53.98 | 70.21 | 60.67 | 76.20 | 89.71 | 90.22 | 78.94 | 76.82 | 60.49 | 63.62 | 73.12 | 71.43 | 61.96 | 73.03 |
| RSDet [64] | ResNet 152 | 90.2 | 83.5 | 53.6 | 70.1 | 64.6 | 79.4 | 67.3 | 91.0 | **88.3** | 82.5 | 64.1 | 68.7 | 62.8 | 69.5 | 66.9 | 73.5 |
| CFC-Net [51] | ResNet 50 | 89.08 | 80.41 | 52.41 | 70.02 | 76.28 | 78.11 | 87.21 | 90.89 | 84.47 | 85.64 | 60.51 | 61.52 | 67.82 | 68.02 | 50.09 | 73.50 |
| R³Det [37] | ResNet 101 | 88.76 | 83.09 | 50.91 | 67.27 | 76.23 | 80.39 | 86.72 | 90.78 | 84.68 | 83.24 | 61.98 | 61.35 | 66.91 | 70.63 | 53.94 | 73.79 |
| SLA [21] | ResNet 50 | 85.23 | 83.78 | 48.89 | 71.65 | 76.43 | 76.80 | 86.83 | 90.62 | 88.17 | 86.88 | 49.67 | 66.13 | 75.34 | 72.11 | 64.88 | 74.89 |
| RDD [65] | ResNet 101 | 89.70 | 84.33 | 46.35 | 68.62 | 73.89 | 73.19 | 86.92 | 90.41 | 86.46 | 84.30 | 64.22 | 64.95 | 73.55 | 72.59 | 73.31 | 75.52 |
| Two-stage |
| FR-O [30] | ResNet 101 | 79.42 | 77.13 | 17.7 | 64.05 | 35.3 | 38.02 | 37.16 | 89.41 | 69.64 | 59.28 | 50.3 | 52.91 | 47.89 | 47.4 | 46.3 | 54.13 |
| RRPN [23] | ResNet 101 | 88.52 | 71.20 | 31.66 | 59.30 | 51.85 | 56.19 | 57.25 | 90.81 | 72.84 | 67.38 | 56.69 | 52.84 | 53.08 | 51.94 | 53.58 | 61.01 |
| FFA [66] | ResNet 101 | 81.36 | 74.30 | 47.70 | 70.32 | 64.89 | 67.82 | 69.98 | 90.76 | 79.06 | 78.20 | 53.64 | 62.90 | 67.02 | 64.17 | 50.23 | 68.16 |
| RADet [53] | ResNeXt 101 | 79.45 | 76.99 | 48.05 | 65.83 | 65.46 | 74.40 | 68.86 | 89.70 | 78.14 | 74.97 | 49.92 | 64.63 | 66.14 | 71.58 | 62.16 | 69.09 |
| RoI Transformer [20] | ResNet 101 | 88.64 | 78.52 | 43.44 | 75.92 | 68.81 | 73.68 | 83.59 | 90.74 | 77.27 | 81.46 | 58.39 | 53.54 | 62.83 | 58.93 | 47.67 | 69.56 |
| CAD-Net [48] | ResNet 101 | 87.8 | 82.4 | 49.4 | 73.5 | 71.1 | 63.5 | 76.7 | 90.9 | 79.2 | 73.3 | 48.4 | 60.9 | 62.0 | 67.0 | 62.2 | 69.9 |
| SCR-Det [54] | ResNet 101 | 89.98 | 80.65 | 52.09 | 68.36 | 68.36 | 60.32 | 72.41 | 90.85 | 87.94 | 86.86 | 65.02 | 66.68 | 66.25 | 68.24 | 65.21 | 72.64 |
| ROSD [50] | ResNet 101 | 88.88 | 82.13 | 52.85 | 69.76 | 78.21 | 77.32 | 87.08 | 90.86 | 86.40 | 82.66 | 56.73 | 65.15 | 74.43 | 68.24 | 63.18 | 74.92 |
| Gliding Vertex [22] | ResNet 101 | 89.64 | 85.00 | 52.26 | 77.34 | 73.01 | 73.14 | 86.82 | 90.74 | 79.02 | 86.81 | 59.55 | 70.91 | 72.94 | 70.86 | 57.32 | 75.02 |
| SAR [57] | ResNet 101 | 89.67 | 79.78 | 54.17 | 68.29 | 71.70 | 77.90 | 84.63 | **90.91** | 88.22 | 87.07 | 60.49 | 66.95 | 75.13 | 70.01 | 64.29 | 75.28 |
| Mask-OBB [38] | ResNeXt 101 | 89.56 | 85.95 | 54.21 | 72.90 | 76.52 | 74.16 | 85.63 | 89.85 | 83.81 | 86.48 | 54.89 | 69.64 | 73.94 | 69.06 | 63.32 | 75.33 |
| APE [67] | ResNet 50 | 89.96 | 83.62 | 53.42 | 76.03 | 74.01 | 77.16 | 79.45 | 90.83 | 87.15 | 84.51 | 67.72 | 60.33 | 74.61 | 71.84 | 65.55 | 75.75 |
| CenterMap-Net [39] | ResNet 101 | 89.83 | 84.41 | 54.60 | 70.25 | 77.66 | 78.32 | 87.19 | 90.66 | 84.89 | 85.27 | 56.46 | 69.23 | 74.13 | 71.56 | 66.06 | 76.03 |
| CSL [55] | ResNet 152 | 90.25 | 85.53 | 54.64 | 75.31 | 70.44 | 73.51 | 77.62 | 90.84 | 86.15 | 86.69 | 69.60 | 68.04 | 73.83 | 71.10 | 68.93 | 76.17 |
| ReDet [68] | ResNet 50 | 88.79 | 82.64 | 53.97 | 74.00 | 78.13 | 84.06 | 88.04 | 90.89 | 87.78 | 85.75 | 61.76 | 60.39 | 75.96 | 68.07 | 63.59 | 76.25 |
| OPLD [69] | ResNet 101 | 89.37 | 85.82 | 54.10 | 79.58 | 75.00 | 75.13 | 86.92 | 90.88 | 86.42 | 86.62 | 62.46 | 68.41 | 73.98 | 68.11 | 63.69 | 76.43 |
| HSP [70] | ResNet 101 | **90.39** | 86.23 | 56.12 | **80.59** | 77.52 | 73.26 | 83.78 | 90.80 | 87.19 | 85.67 | 69.08 | **72.02** | 76.98 | 72.50 | 67.96 | 78.01 |
| Anchor-free |
| CenterNet-O [26] | Hourglass 104 | 89.02 | 69.71 | 37.62 | 63.42 | 65.23 | 63.74 | 77.28 | 90.51 | 79.24 | 77.93 | 44.83 | 54.64 | 55.93 | 61.11 | 45.71 | 65.04 |
| Axis Learning [41] | ResNet 101 | 79.53 | 77.15 | 38.59 | 61.15 | 67.53 | 70.49 | 76.30 | 89.66 | 79.07 | 83.53 | 47.27 | 61.01 | 56.28 | 66.06 | 36.05 | 65.98 |
| P-RSDet [62] | ResNet 101 | 88.58 | 77.84 | 50.44 | 69.29 | 71.10 | 75.79 | 78.66 | 90.88 | 80.10 | 81.71 | 57.92 | 63.03 | 66.30 | 69.70 | 63.13 | 72.30 |
| BBAVectors [59] | ResNet 101 | 88.35 | 79.96 | 50.69 | 62.18 | 78.43 | 78.98 | 87.94 | 90.85 | 83.58 | 84.35 | 54.13 | 60.24 | 65.22 | 64.28 | 55.70 | 72.32 |
| O²-Det [56] | Hourglass 104 | 89.3 | 83.3 | 50.1 | 72.1 | 71.1 | 75.6 | 78.7 | 90.9 | 79.9 | 82.9 | 60.2 | 60.0 | 64.6 | 68.9 | 65.7 | 72.8 |
| PolarDet [61] | ResNet 50 | 89.73 | **87.05** | 45.30 | 63.32 | 78.44 | 76.65 | 87.13 | 90.79 | 80.58 | 85.89 | 60.97 | 67.94 | 68.20 | 74.63 | 68.67 | 75.02 |
| AOPG [31] | ResNet 101 | 89.14 | 82.74 | 51.87 | 69.28 | 77.65 | 82.42 | 88.08 | 90.89 | 86.26 | 85.13 | 60.60 | 66.30 | 74.05 | 67.76 | 58.77 | 75.39 |
| CBDANet [52] | DLA 34 | 89.17 | 85.92 | 50.28 | 65.02 | 77.72 | 82.32 | 87.89 | 90.48 | 86.47 | 85.90 | 66.85 | 66.48 | 67.41 | 71.33 | 62.89 | 75.74 |
| CFA [42] | ResNet 152 | 89.08 | 83.20 | 54.37 | 66.87 | **81.23** | 80.96 | 87.17 | 90.21 | 84.32 | 86.09 | 52.34 | 69.94 | 75.52 | **80.76** | 67.96 | 76.67 |
| Proposed Method | ResNet 101 | 89.23 | 84.50 | 52.90 | 76.93 | 78.51 | 76.93 | 87.40 | 90.89 | 87.42 | 84.66 | 64.40 | 63.97 | 75.01 | 73.39 | 62.37 | 76.57 |
| Proposed Method * | ResNet 101 | 90.20 | 84.94 | **61.04** | 79.66 | 79.73 | **84.37** | **88.78** | 90.88 | 86.16 | **87.66** | **71.85** | 70.40 | **81.37** | 79.71 | **73.51** | **80.68** |

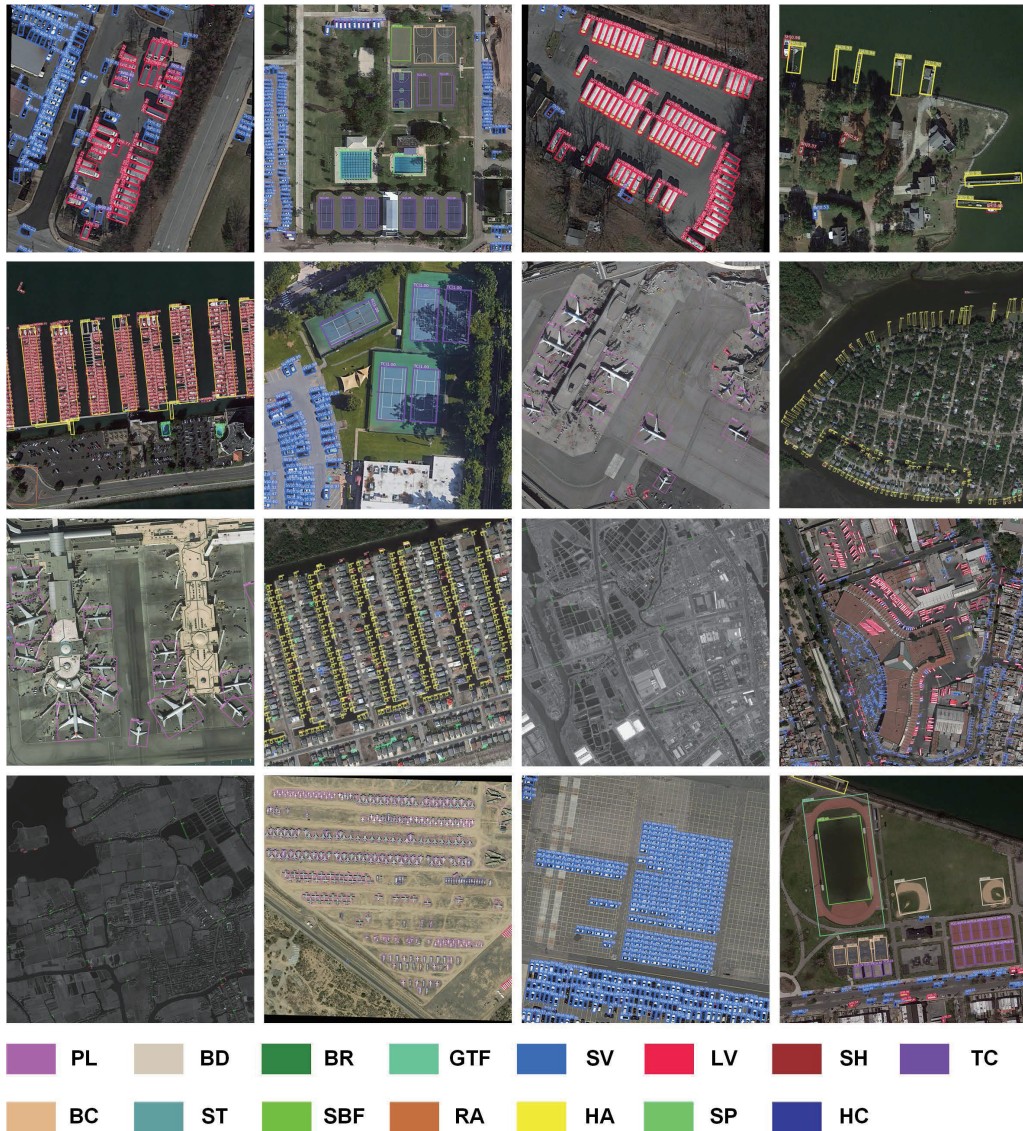

| | PL | | BD | | BR | | GTF | | SV | | LV | | SH | | TC |
| | BC | | ST | | SBF | | RA | | HA | | SP | | HC |

**Figure 7.** Depictions of the detection results on the DOTA dataset test set. We use bounding boxes of different colors to represent different categories.

### 3.3.2. Results on DIOR-R

DIOR-R is a new oriented object detection dataset, so we retrained and tested several advanced methods to conduct fair performance comparisons. As shown in Table 2, Faster RCNN OBB [30], as the baseline two-stage oriented method, and RetinaNet OBB [19], as the baseline single-stage oriented method, achieved 57.14% and 55.92% mAP, respectively. As the advanced methods, RoI Transformer [20] and Gliding Vertex [22] achieved 65.93% and 61.81% mAP, respectively. AOPG [31], as the baseline method in the DIOR-R dataset, achieved 64.41% mAP. Our model achieved 65.80% mAP with ResNet 50 [12] as the backbone and 67.15% mAP with ResNet 101 [12] as the backbone. The detection results are depicted in Figure 8.

**Table 2.** Comparisons with state-of-the-art methods on DIOR-R dataset test set. **Bold** denotes the best detection results.

| Method | Backbone | APL | APO | BF | BC | BR | CH | DAM | ETS | ESA | GF | GTF | HA | OP | SH | STA | STO | TC | TS | VE | WM | mAP |
|---|---|---|---|---|---|---|---|---|---|---|---|---|---|---|---|---|---|---|---|---|---|---|
| RetinaNet-O [19] | ResNet 101 | 64.20 | 21.97 | 73.99 | 86.76 | 17.57 | 72.62 | 72.36 | 47.22 | 22.08 | 77.90 | 76.60 | 36.61 | 30.94 | 74.97 | 63.35 | 49.21 | 83.44 | 44.93 | 37.53 | 64.18 | 55.92 |
| FR-O [30] | ResNet 101 | 61.33 | 14.73 | 71.47 | 86.46 | 19.86 | 72.24 | 59.78 | 55.98 | 19.72 | 77.08 | 81.47 | 39.21 | 33.30 | 78.78 | 70.05 | 61.85 | 81.31 | 53.44 | 39.90 | 64.81 | 57.14 |
| Gliding Vertex [22] | ResNet 101 | 61.58 | 36.02 | 71.61 | 86.87 | 33.48 | 72.37 | 72.85 | 64.62 | 25.78 | 76.03 | 81.81 | 42.41 | 47.25 | 80.57 | 69.63 | 61.98 | 86.74 | 58.20 | 41.87 | 64.48 | 61.81 |
| AOPG [31] | ResNet 50 | 62.39 | 37.79 | 71.62 | 87.63 | 40.90 | 72.47 | 31.08 | 65.42 | 77.99 | 73.20 | 81.94 | 42.32 | 54.45 | 81.17 | 72.69 | 71.31 | 81.49 | 60.04 | 52.38 | 69.99 | 64.41 |
| RoI Trans [20] | ResNet 101 | 61.54 | 45.46 | 71.90 | 87.48 | **41.43** | 72.67 | 78.67 | **67.17** | **38.26** | **81.83** | **83.40** | **48.94** | 55.61 | 81.18 | **75.06** | 62.63 | 88.36 | **63.09** | 47.80 | 66.10 | 65.93 |
| Proposed Method | ResNet 50 | **68.26** | 38.34 | 77.35 | 88.10 | 40.68 | 72.48 | 78.90 | 62.52 | 30.64 | 73.51 | 81.32 | 45.51 | **55.78** | **88.74** | 71.24 | 71.12 | **88.60** | 59.74 | **52.95** | **70.30** | 65.80 |
| Proposed Method | ResNet 101 | 61.65 | **47.58** | **77.59** | **88.39** | 40.98 | 72.55 | **81.90** | 63.76 | 38.17 | 79.49 | 81.82 | 45.39 | 54.94 | 88.67 | 73.48 | **75.75** | 87.69 | 61.69 | 52.43 | 69.00 | **67.15** |

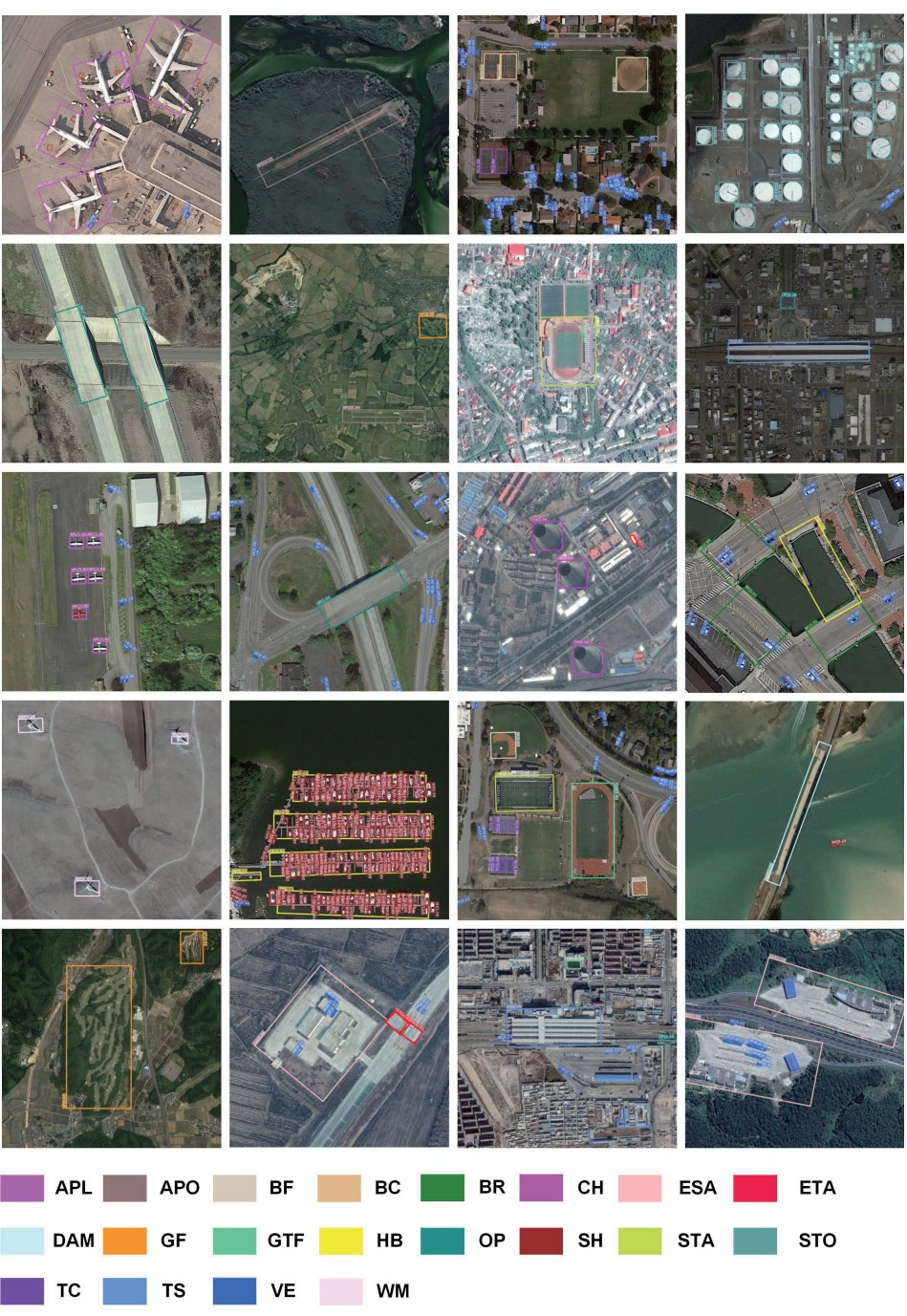

**Figure 8.** Depictions of the detection results on the DIOR-R dataset test set. We use bounding boxes of different colors to represent different categories.

### 3.3.3. Results on HRSC2016

The HRSC2016 dataset contains many densely packed ship instances with arbitrary orientation and large aspect ratios. Table 3 shows the results of our comparison of the proposed method with several state-of-the-art methods. Our model achieved 89.96% mAP with ResNet 50 as the backbone and 90.45% mAP with ResNet 101 as the backbone, which shows the effectiveness of dealing with such objects. As shown in Figure 9, our model accurately detects ships in complex remote sensing images.

**Table 3.** Comparisons with other methods on HRSC2016 dataset test set. **Bold** denotes the best detection results.

| Method | Backbone | Image Size | mAP |
|---|---|---|---|
| Axis Learning [41] | ResNet 101 | 800 × 800 | 78.15 |
| SLA [21] | ResNet 50 | 768 × 768 | 87.14 |
| SAR [57] | ResNet 101 | 896 × 896 | 88.11 |
| Gliding Vertex [22] | ResNet 101 | - | 88.2 |
| OPLD [69] | ResNet 50 | 1024 × 1333 | 88.44 |
| BBAVectors [59] | ResNet 101 | 608 × 608 | 88.6 |
| DAL [63] | ResNet 101 | 800 × 800 | 88.6 |
| ProjBB-R [58] | ResNet 101 | 800 × 800 | 89.41 |
| CSL [55] | ResNet 152 | - | 89.62 |
| CFC-Net [51] | ResNet 101 | 800 × 800 | 89.7 |
| ROSD [50] | ResNet 101 | 1000 × 800 | 90.08 |
| PolarDet [61] | ResNet 50 | 800 × 800 | 90.13 |
| AOPG [31] | ResNet 101 | 800 × 1333 | 90.34 |
| ReDet [68] | ResNet 50 | 800 × 512 | 90.46 |
| CBDANet [52] | DLA 34 | 512 × 512 | **90.5** |
| Proposed Method | ResNet 50 | 800 × 1333 | 89.96 |
| Proposed Method | ResNet 101 | 800 × 1333 | 90.45 |

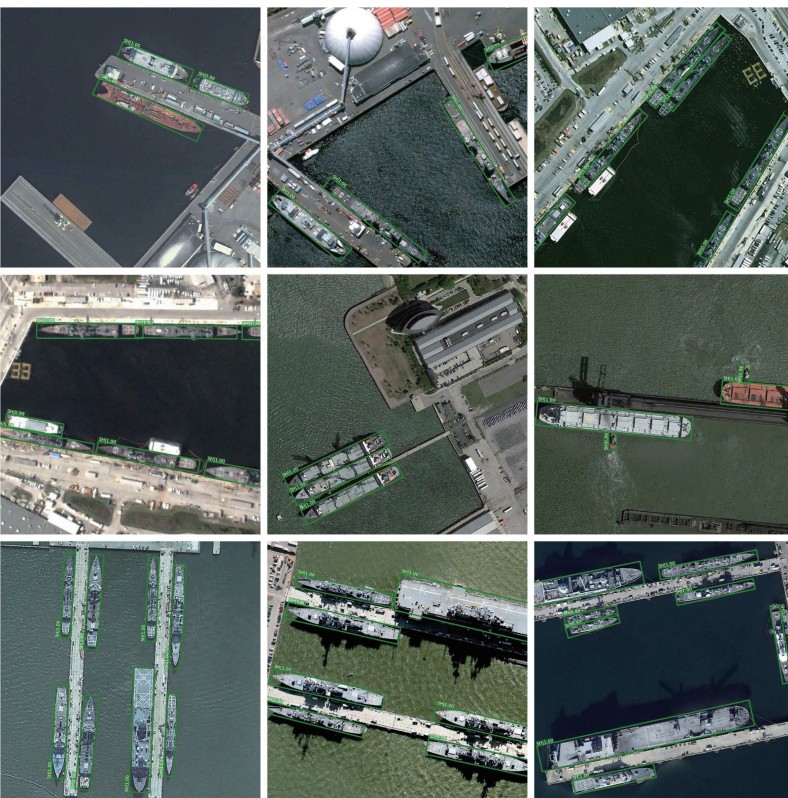

**Figure 9.** Depictions of the detection results on the HRSC2016 dataset test set.

## 4. Discussion

### 4.1. Ablation Study

To verify the effectiveness of the proposed method, we conducted ablation studies on the DOTA dataset test set. We used the RoI Transformer [20] with ResNet 101 [12] as the baseline in the experiments. It can be seen from the first row in Table 4 that the baseline method achieved 69.56% mAP, and from the fourth row that the proposed method with both CCA-FPN and AFO-RPN modules achieved a significant improvement of 7.01% mAP. Some visual comparison examples are shown in Figure 10.

**Table 4.** Ablation study of proposed modules on DOTA dataset test set.

| Method | CCA-FPN | AFO-RPN | PL | BD | BR | GTF | SV | LV | SH | TC | BC | ST | SBF | RA | HA | SP | HC | mAP(%) |
|---|---|---|---|---|---|---|---|---|---|---|---|---|---|---|---|---|---|---|
| Baseline [20] | - | - | 88.64 | 78.52 | 43.44 | 75.92 | 68.81 | 73.68 | 83.59 | 90.74 | 77.27 | 81.46 | 58.39 | 53.54 | 62.83 | 58.93 | 47.67 | 69.56 |
| Proposed Method | ✓ | - | 88.59 | 81.60 | 52.27 | 68.19 | 78.02 | 73.69 | 86.64 | 90.74 | 82.97 | 85.12 | 56.31 | 65.38 | 69.66 | 68.50 | 56.75 | 73.63 (+4.07) |
| | - | ✓ | 88.88 | 84.06 | 52.13 | 69.55 | 70.96 | 76.59 | 79.52 | 90.87 | 87.23 | 86.19 | 56.14 | 65.35 | 66.96 | 72.08 | 64.20 | 74.05 (+4.49) |
| | ✓ | ✓ | 89.23 | 84.50 | 52.90 | 76.93 | 78.51 | 76.93 | 87.40 | 90.89 | 87.42 | 84.66 | 64.40 | 63.97 | 75.01 | 73.39 | 62.37 | 76.57 (+7.01 ) |

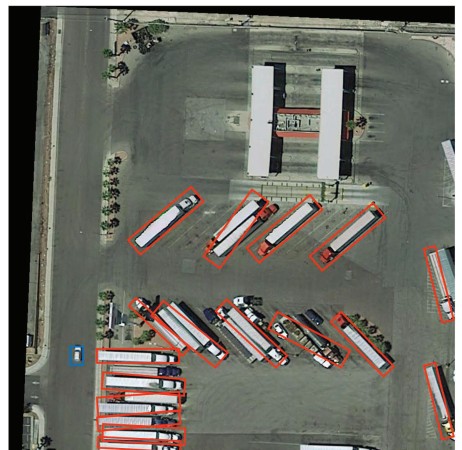
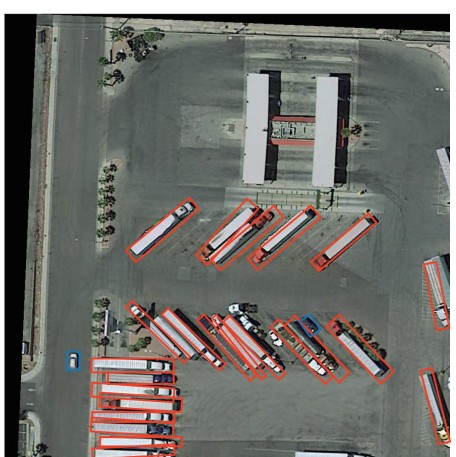
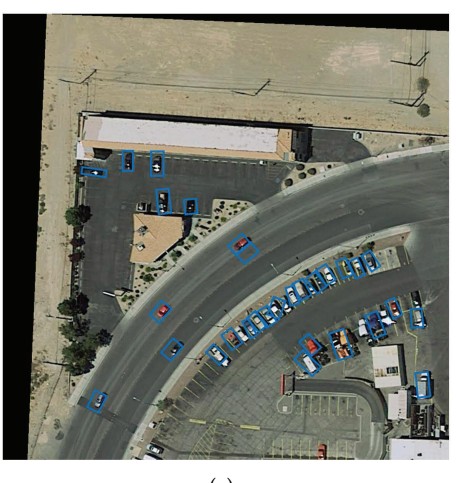
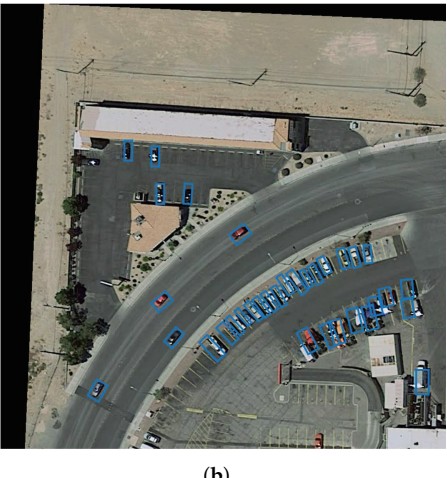

(**a**)    (**b**)

**Figure 10.** Depictions of the detection results on the DOTA dataset test set. (**a**) Baseline [20]. (**b**) Proposed method.

#### 4.1.1. Effect of the Proposed AFO-RPN

The third row of Table 4 shows 4.49% increases in terms of mAP with the AFO-RPN module. The proposed AFO-RPN is designed to generate high quality–oriented proposals without placing excessive fix-shaped anchors on the feature map. The accuracy for hard instance categories such as BD, BR, LV, BC, and HC increased by 5.54%, 8.69%, 2.91%, 9.96%,

and 16.53% in terms of mAP, respectively. However, the accuracy for some categories such as GTF, SH, SBF decreased by 6.37%, 4.07%, and 2.25% in terms of mAP. The reason is that AFO-RPN is keypoint-based anchor-free method and it could face severe appearance ambiguity problems with backgrounds or other categories, as shown in Figure 2. The results prove the weakness of the anchor-free method

### 4.1.2. Effect of the CCA-FPN

The second row of Table 4 shows 4.07% increases in terms of mAP with CCA-FPN module. CCA-FPN is designed to enhance the feature representation of each pixel by capturing the contextual information. The accuracy for some hard instance categories, such as BR, SV, SH, BC, and RA, increased by 8.83%, 9.21%, 3.05%, 5.7%, and 11.84% in terms of mAP, respectively. It can be seen from the last two rows in Table 4, the performances for GTF, SH, SBF increased by 7.38%, 7.88%, 8.26% in terms of mAP, respectively. It shows contextual information is useful to enhance the representation of each point on the feature map.

We also compared the model's parameters (Params) and calculations (FLOPs) of the proposed method with baseline. The sizes of the input image are $800 \times 800$ pixels. The smaller Params and FLOPs, the higher the efficiency and the shorter inference time of the detector. The second row of Table 5 shows that the proposed method with AFO-RPN module has fewer parameters and low computational complexity. However, the third row of Table 5 shows that the CCA-FPN module brings huge parameters and a high computational burden.

**Table 5.** Evaluation results with the parameters and computational complexity.

| Method | CCA-FPN | AFO-RPN | Params(M) | FLOPs(G) |
|:---:|:---:|:---:|:---:|:---:|
| Baseline [20] | - | - | 55.13 | 148.38 |
| Proposed Method | - | ✓ | 41.73 | 134.38 |
|  | ✓ | ✓ | 65.66 | 376.99 |

### 4.1.3. Effect of the Proposed Polar Representation of OBB

To explore the impacts of different OBB representation methods, we compared the proposed polar representation method with two commonly used Cartesian system representation methods—angle-based representation $(x, y, w, h, \theta)$ and vertex-based representation $(x_1, y_1, x_2, y_2, x_3, y_3, x_4, y_4)$—on the DOTA, DIOR-R, and HRSC2016 datasets. As shown in Table 6, the proposed polar representation method achieved a significant increase over the Cartesian system representation methods in all three datasets.

**Table 6.** Ablation study of proposed polar representation method of OBB.

| Cartesian System | Polar System | DOTA mAP(%) | DIOR-R mAP(%) | HRSC2016 mAP(%) |
|:---:|:---:|:---|:---|:---|
| $(x, y, w, h, \theta)$ | - | 73.84 | 64.81 | 88.12 |
| $(x_1, y_1, x_2, y_2, x_3, y_3, x_4, y_4)$ | - | 72.58 | 63.48 | 84.84 |
| - | $(x, y, \rho, \gamma, \varphi)$ | 76.57 | 67.15 | 90.45 |

### 4.2. Limitations

As shown in Table 4, the utilization of the proposed AFO-RPN module improves the performance on many categories but degrades the performance on several categories. To solve this problem, we apply an attention module Criss-Cross Attention into FPN to enhance the feature representation by exploiting the contextual information. The proposed method with both CCA-FPN and AFO-RPN modules achieved a significant improvement while encountering another problem of calculation complexity, as shown in Table 5. This is a problem to be solved in future work.

## 5. Conclusions

In this paper, we analyzed the drawbacks of the mainstream anchor-based methods and found that both horizontal anchors and oriented anchors will hinder the further improvement of the oriented object detection results. To address this, we propose a two-stage coarse-to-fine oriented detector. The proposed method has the following novel features: (1) the proposed AFO-RPN, which generates high-quality oriented proposals without enormous predefined anchors; (2) the CCA-FPN, which enhances the feature representation of each pixel by capturing the contextual information; and (3) a new representation method of the OBB in the polar coordinates system, which slightly improves the detection performance. Extensive ablation studies have shown the superiority of the proposed modules. We achieved mAPs of 80.68% on the DOTA dataset, 67.15% on the DIOR-R dataset, and 90.45% on the HRSC2016 dataset, demonstrating that our method can achieve promising performance compared with the state-of-the-art methods.

However, despite the good performance, our method increased the parameters and computation cost. We will focus on improving the method and reducing the calculation burden in our future work.

**Author Contributions:** Conceptualization, J.L. and Y.T.; methodology, J.L.; software, J.L.; validation, J.L., Y.X. and Z.Z.; formal analysis, J.L. and Y.T.; investigation, Y.X. and Z.Z.; resources, Y.T. and Y.X.; data curation, J.L., Y.X. and Z.Z.; writing—original draft preparation, J.L.; writing—review and editing, Y.T.; visualization, J.L. and Z.Z.; supervision, Y.T.; project administration, J.L. All authors have read and agreed to the published version of the manuscript.

**Funding:** This research received no external funding.

**Institutional Review Board Statement:** Not applicable.

**Informed Consent Statement:** Not applicable.

**Data Availability Statement:** The datasets used in this study are available on request from the corresponding author.

**Conflicts of Interest:** The authors declare no conflict of interest.

## Abbreviations

The following abbreviations are used in this manuscript:

| | |
|---|---|
| RSI | Remote Sensing Image |
| DCNN | Deep Convolutional Neural Network |
| RSI | Remote Sensing Image |
| HBB | Horizontal Bounding Box |
| OBB | Oriented Bounding Box |
| RPN | Region Proposal Network |
| RoI | Region of Interest |
| FPN | Feature Pyramid Network |
| mAP | mean Average Precision |
| AFO-RPN | Anchor-Free Oriented Region Proposal Network |
| CCA-FPN | Criss-Cross Attention Feature Pyramid Network |

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
