# Peer review of "Oriented Object Detection in Remote Sensing Images with Anchor-Free Oriented Region Proposal Network"

_remotesensing, doi:10.3390/rs14051246_

Round 1
Reviewer 1 Report
This paper proposes a two-stage anchor-free network for oriented object detection. Specifically, the authors apply Criss-Cross Attention into the task of oriented object detection. Next, the AFO-RPN is proposed to generate oriented proposals. Moreover, a new representation form of oriented box is proposed based on polar coordinate system. Finally, the Oriented R-CNN is used to refine the final results. Extensive experiments on three datasets validate the superiority of the proposed method over state-of-the-art methods. The proposed method is novel to some extent, experimental results are convincing. I think this manuscript requires a minor revision.
(1) This paper successfully applies Criss-Cross Attention module into the task of oriented object detection and achieves considerable improvement.
(2) The proposed AFO-RPN is novel and is the key innovation in this paper, it seems that it effectively combines the proposed polar representation with anchor free regional proposal network.
(3) The proposed polar representation is interesting. However, it seems that similar ways of representing OBB have been proposed in PolarDet [1]. I would like the authors to provide more details about the difference between the proposed form and that in PolarDet.
(4) Also, the motivation of introducing the polar coordinate is not clear enough. In other words, what is the specific advantage of the proposed forms over other forms (i.e. angle-based and vertex-based) and similar forms in PolarDet. It may be better if more analysis can be added into the paper.
(5) Figures in this paper are not well-organized. Most figures are of low resolution. Fig.3 and Fig.7 are too large and take up too much space.
(6) For visualization results, they are not clear and it is hard to find out the superiority of the proposed method. I would like the authors to provide comparison results between different methods.
(7) The proposed method achieves state-of-the-art result on several challenging datasets, so that it can be considered solid and encouraging work. I encourage that the authors provide open-source code if possible.
[1] Learning Polar Encodings for Arbitrary-Oriented Ship Detection in SAR Images. IEEE J-STARS, 2021.
Reviewer 2 Report
This manuscript addresses 2D object detection problem in remote sensing image data. The enhanced feature representation of the input image is first acquired using an attention module, and feeding the representation into an anchor-free RPN, oriented bounding boxes of the objects of interest are hypothesised.
The introduction provides sufficient background for the problem, the paper is well-structured, and the methodology is adequately described. The arguments given in the introduction are advocated in the experimental work section, however, elaborating discussions will help to improve the manuscript:
1) In the introduction, it is given that RRPN [21] uses high number of anchors and thus giving rise an increase in the computation burden. The authors can consider depicting comparative numbers in the experimental works section so as one can see the efficiency of the presented method time-wise.
2) The scales used for training: The authors, when augmenting their training data with different scales, choose 0.5, 1, and 1.5 ratios. How do you determine these numbers, empirically or according to the study in [18]? If it is empiric, does the model work on the instances out of this range? The authors can further discuss scale, resolution.
3) According to Table 1, bare utilization of AFO-RPN atop of baseline degrades the performance on several categories, such as GTF, SH, SBF. Why? Further discussion would clarify this point, particularly for the SH object, since the last dataset (HRSC2016) is devoted to SH category.
4) The authors can lastly spell check the whole paper.
Reviewer 3 Report
This manuscript presents an anchor-free rotated object detection method by proposing AFO-RPN in two stage framework.
The proposed scheme (AFO-RPN and polar representation) showed upgraded performance compared to several benchmark DB.
Followings should be revised to improve your paper.
- You have to clarify your idea and previous work.
For example, 4.3.1 Effect of the Proposed CCA-FPN -> not propsed
CCA and FPN are previous works.
2. Figure 5. You have to cite the image reference and get the copyright for publication. Or, remove the figure.
3. Others: check typos and styles.
Reviewer 4 Report
General evolution
I read this paper carefully. I n my opinion the methods and techniques is sufficient and interesting, however paper requires reorganizing in terms of the structure. Sections are mixed up and I think the authors failed to provide a good representation. I have provided several comments and I hope would support them to improve it essentially.
Oriented object, is a bit strange! Everybody knows it as object oriented or object based image analysis (OBIA), please reconsider the term, I would suggest OBIA
The abstract section has to be revised, it is just a method description and there is no indication regarding data, results, and conclusion statement.
The introduction section is fine but it lacks the new progress in OBIA which is achieved by remote sensing society already. Right now, the integration of OBIA with machine learning and deep learning techniques developed very well, for example, see the following articles: I would really suggest the author improve this section and their justification regarding the new approach of OBIA and its integration with machine learning techniques.
line 88-99: it is clear that it is your objective in this study, but this format is a bit strange to me and for the scientific community, I believe. This is a research paper rather than a proposal, so you should combine these statements in a nice paragraph and also highlight the research objective, research gap, and the state of art in your study rather than surmising them point by point which makes the paper as proposal.
The paper needs essential works regarding the structure, it really has to be reorganized in the normal research paper by having the method, results, discussion sections. All sections are now combined and makes really confusing what are the results, there is no even discussion regarding the efficiency of proposed methods? The validation is missing, how author could reclaim the approach is efficient without sufficient validation?
The conclusion is not really a conclusion and mixed up with results and methodology explanation, it has to be focused what are the research significant for the remote sensing scarcity in terms of the proposing method and etc. perhaps possible applications for future studies might be also addressed.
Round 2
Reviewer 4 Report
Thanks for the comprehensive revise